# Spatial Data Reconstruction via ADMM and Spatial Spline Regression †

**Bang Liu** [1,]*** , **Borislav Mavrin** [2], **Linglong Kong** [2] **and Di Niu** [1]

1   Electrical and Computer Engineering, University of Alberta, 9211-116 Street NW, Edmonton, AB T6G 1H9, Canada; dniu@ualberta.ca
2   Mathematical and Statistical Sciences, University of Alberta, 632 Central Academic Building, Edmonton, AB T6G 2G1, Canada; mavrin@ualberta.ca (B.M.); lkong@ualberta.ca (L.K.)
*   Correspondence: bang3@ualberta.ca
†   This paper is an extended version of our paper published in the 2017 IEEE International Conference on Data Mining (ICDM), New Orleans, LA, USA, 18–21 November 2017.

**Abstract:** Reconstructing fine-grained spatial densities from coarse-grained measurements, namely the aggregate observations recorded for each subregion in the spatial field of interest, is a critical problem in many real world applications. In this paper, we propose a novel Constrained Spatial Smoothing (CSS) approach for the problem of spatial data reconstruction. We observe that local continuity exists in many types of spatial data. Based on this observation, our approach performs sparse recovery via a finite element method, while in the meantime enforcing the aggregated observation constraints through an innovative use of the Alternating Direction Method of Multipliers (ADMM) algorithm framework. Furthermore, our approach is able to incorporate external information as a regression add-on to further enhance recovery performance. To evaluate our approach, we study the problem of reconstructing the spatial distribution of cellphone traffic volumes based on aggregate volumes recorded at sparsely scattered base stations. We perform extensive experiments based on a large dataset of Call Detail Records and a geographical and demographical attribute dataset from the city of Milan, and compare our approach with other methods such as Spatial Spline Regression. The evaluation results show that our approach significantly outperforms various baseline approaches. This proves that jointly modeling the underlying spatial continuity and the local features that characterize the heterogeneity of different locations can help improve the performance of spatial recovery.

**Keywords:** spatial sparse recovery; constrained spatial smoothing; spatial spline regression; alternating direction method of multipliers

## 1. Introduction

The problem of reconstructing fine-grained spatial data from its coarse-grained aggregate observations of each subregions lies in the core of many real world applications. For example, the reconstruction of fine-grained spatial distribution of cell phone activities is of particular interest to telecommunication and information technology companies, where the recovered data can be used for device installation, capacity planning, the study of urban ecology [1–3], population density estimation [4–6], and human mobility prediction [7–11]. However, the companies may only have access to the aggregate mobile traffic volumes on each base station, as either privacy issues or additional technical overhead is involved to get fine-grained spatial data of users. Similarly, it is also highly valuable if we can infer the spatial distribution of population (e.g., the population vote for a certain party) densities based on the total population recorded at polling stations that sparsely scattered

at different subregions. Internet media providers or retailers, such as Google, Tencent, Amazon, Facebook, etc., may want to recover a fine-grained geographical distribution of their users based on the aggregated user counts observed at different points of presence (PoPs) or data centers. Note that, in all the above-mentioned cases, it is impossible or not allowed to track the position of each individual due to either privacy concerns or technical overhead. Therefore, reconstructing the spatial data from coarse aggregation will be highly useful in such cases.

In this paper, we study such spatial sparse recovery problem, that is, to infer the fine-grained distribution of certain spatial data in a region given the aggregate observations recorded for each of its subregions. However, it is an extremely challenging problem and has seldom been studied. A straightforward idea is assuming the density is uniformly distributed within each subregion. Based on the based on the obtained aggregate observation, we can calculate a patched piece-wise constant estimation for each subregion. However, the densities estimated by this method will jump between neighboring subregions and disregard the local continuity or similarity of the studied spatial distribution across subregion boundaries. In addition, the piece-wise constant spatial field given by this approach provides little value for applications such as hot spot discovery. Many spatial data presents local continuity, e.g., Internet activity or cell phone activity. This is because the data often highly depend on underlying factors which are usually smoothly changing, like area functionality, urban geographical features, population density and so on. To exploit the smoothness, we may utilize spatial smoothing techniques such as Thin Plate Splines [12], Soap film smoothing [13], Spline smoothing [14], Bivariate Spline Regression [15], or Spatial Spline Regression [16] developed in statistics to smoothen the patched estimation. However, nearly all existing spatial smoothing techniques [12–16] are designed to recover a spatial field of densities according to sampled observations, e.g., reconstruct a spatial field of temperatures based on the temperature records at some sample points. In contrast, our problem needs to recovery a spatial field based on coarse-grained aggregate observations. Therefore, existing spatial smoothing techniques are not directly applicable to our new problem. Without modification, these smoothing techniques will violate the necessary constraint that the estimated spatial data in each subregion must sum up to its corresponding aggregate observation in the first place, leading to systematic errors.

To overcome the difficulties mentioned above, in this paper, we propose a new technique named Constrained Spatial Smoothing (CSS) for the problem of spatial data reconstruction. Specifically, given a region, we aim to reconstruct a spatial field of densities over that region based on observed aggregate values in patched subregions. Our approach penalizes the "roughness" of the reconstructed spatial field subject to the constraint that the aggregation of discretized values of the spatial field in each patched subregion equals the aggregate observation made in that subregion. It is distinct from previous spatial smoothing techniques due to the additional constraint in our problem. We propose an Alternating Direction Method of Multipliers (ADMM) [17,18] algorithm to decouple the problem into the alternated minimizations of a quadratic program (QP) [19] subproblem and a spatial smoothing subproblem, where we use the QP to iteratively enforce the observation constraints, while solving the spatial smoothing subproblem with a recently proposed finite element technique called Spatial Spline Regression (SSR) [16]. In addition, our approach not only leverages the intrinsic smoothness from local continuity to reconstruct a spatial field, but is also able to incorporate additional external information, such as the number of schools, number of bus stops, population, etc., in the underlying geographical region as a regression add-on component to further enhance recovery performance. Last but not least, our algorithm can be applied to a variety of sparse recovery problem where intrinsic smoothness exists.

Another important contribution of the paper is that we conduct extensive evaluation to compare our proposed algorithms with a variety of baseline methods. In our evaluation, we are trying to reconstruct the mobile phone activity distributions in Milan, Italy from base station observations. The Telecom Italia Big Data Challenge dataset is a multi-source dataset that contains a variety of informations, including aggregation of telecommunication activities, news, social networks, weather, and electricity data from the city of Milan. With the important information about human activities

contained in the dataset, especially the cellphone activity records, researchers utilized the data to study different problems, such as modeling human mobility patterns [20–22], population density estimation [4,5], models the spread of diseases [23,24], modeling city structure [3] and city ecology [2], etc. Specifically, our evaluation is based on the Milan Call Detail Records (CDR) dataset, a part of the Telecom Italia Big Data Challenge dataset [25] which contains the phone call and Short Message Service (SMS) activity records of two months in each grid square of 235 m × 235 m in the city of Milan, Italy.

Given the Milan Call Detail Records (CDR) dataset, we consider a region that consists of 2726 grid squares in an irregularly bounded region in the city of Milan. To stress-test the algorithm performance, we assume we only know the aggregate phone activities observed on 100 or 200 base stations and aim to recover the entire spatial field of phone activities. We also use another geographical attribute dataset available from the Municipality of Milan's Open Data website [1] as the additional external attribute data to improve performance. Extensive evaluation shows that our proposed approach achieves significant improvement, compared to various state-of-the-art baseline methods, including the spatial spline regression (SSR) [16] approach. Our technique can recover the fine-grained cell phone activity distribution of 2726 data points only from 200 data points of base stations, with a mean absolute percentage error of 0.309, representing a 26.3% improvement from the SSR baseline scheme.

The remainder of this paper is organized as follows. In Section 2, we formulate the problem of spatial field reconstruction from coarse aggregate observations. In Section 3, we describe existing solutions, including a state-of-the-art Spatial Spline Regression (SSR) technique for spatial smoothing. In Section 4, we propose our Constrained Spatial Smoothing method which respects both the local continuity in the spatial field and the aggregation constraints at the same time. In Section 5, we conduct extensive evaluation in comparison with various other methods through a solid and extensive case study of cell phone activity density estimation in the city of Milan. We discuss related literature in Section 6 and conclude the paper in Section 7.

## 2. Problem Formulation

In this section, we formally introduce the problem of spatial field reconstruction from coarse aggregations observed at sparse scattered points in that field. Our problem can be formulated as a new type of sparse recovery problems. To ease the presentation, we may use cell phone activity recovery as an example.

Let $\Omega \subset \mathbb{R}^2$ denote an irregularly bounded domain, which is the entire region of interest in our problem. Usually, it excludes the uninhabited areas such as hills, ocean coasts, rivers, and so on. Suppose $f(\mathbf{p})$ is a real-valued function that represents certain spatial densities field (e.g., cell phone activities), where $\mathbf{p} = (x, y) \in \Omega$ denotes different geographical positions in $\Omega$. Let $B = \{B_1, \ldots, B_m\}$ denote $m$ observation points (e.g., base stations) that scattered in $\Omega$. Each point $B_i$ is located in a position $\mathbf{p}_{B_i} \in \Omega$ and in charge of a subregion $\Omega_{B_i}$. In our problem, we are given the *aggregated volume* $z_i$ in $\Omega_{B_i}$ that $B_i$ is in charge of. Our goal is to reconstruct the spatial field $f(\mathbf{p})$ based on the observed aggregated volumes $z_i$.

To give an instance, consider the problem of recover cell phone activity distribution. In this case, each user will connect to a base station (cell tower) that is closest to his/her cell phone. Therefore, we can observe the aggregated volume for each base station

$$z_i = \int_{\Omega_{B_i}} f(\mathbf{p}) \, d\mathbf{p}, \quad i = 1, \ldots, m,$$

where $\Omega_{B_i}$ denotes the subregion that $B_i$ is in charge of, and is given by

$$\Omega_{B_i} = \{\mathbf{p} \in \Omega : \|\mathbf{p} - \mathbf{p}_{B_i}\| < \|\mathbf{p} - \mathbf{p}_{B_{i'}}\|, \forall B_{i'} \in B, \ i' \neq i\}.$$

Given the aggregated activity volumes $z_1, \ldots, z_m$ recorded on $m$ base stations, our goal is to reconstruct the entire cell phone activity densities distribution $f$, which is a spatial field in the domain

$\Omega$. We may call $z_1, \ldots, z_m$ base station volumes in this case. However, reconstructing a continuous spatial field is almost computationally infeasible as a personal computer can not handle the continuous nature of $\Omega_{B_i}$.

In reality, we only need to recover $f$ to a certain granularity required by the operator (e.g., 235 m $\times$ 235 m squares in the dataset provided by Telecom Italia Mobile). To fix notations, suppose $\Omega$ is discretized into $n$ small grid squares $\mathbf{p}_1, \ldots, \mathbf{p}_n$, where $\mathbf{p}_j = (x_j, y_j) \in \Omega, j = 1, \ldots, n$ are the center positions of each square $j$ in $\Omega$. We can assume the area of each square is $\Delta = 1$ without loss of generality. In addition, the number of aggregate observations is much smaller than the total number of squares to be reconstructed, therefore we have $m \ll n$.

After domain discretization, we can get the aggregate volume on each base station $B_i$ by

$$z_i = \sum_{\mathbf{p}_j \in \Omega_{B_i}} f(\mathbf{p}_j) \cdot \Delta, \quad i = 1, \ldots, m, \tag{1}$$

where the subregion that $B_i$ represents is given by

$$\Omega_{B_i} = \{\mathbf{p}_j : 1 \le j \le n, \|\mathbf{p}_j - \mathbf{p}_{B_i}\| < \|\mathbf{p}_j - \mathbf{p}_{B_{i'}}\|, \forall i' \ne i\}. \tag{2}$$

Therefore, our goal is to reconstruct the underlying spatial field $f$, and especially the activity densities

$$\mathbf{f} := (f(\mathbf{p}_1), \ldots, f(\mathbf{p}_n))^{\mathsf{T}}$$

in all $n$ grid squares if the desired granularity is on a per-square level, with only access to the aggregated observations $z_i$ in Label (1).

The problem defined above is broadly applicable to characterize a variety of applications other than the recovery of cell phone activity density distribution, e.g., inferring a fine-grained geographical user distribution for a certain app or website based on aggregated user counts collected at sparsely distributed Presence of Points (PoPs) or data centers, and recovering the voter distribution for a certain party based on aggregate voting statistics at different polling stations. The nonessential difference is that the definition of subregion $\Omega_{B_i}$, from which volume $z_i$ is aggregated, is different for each specific application.

*Constrained Spatial Smoothing Problem*

Denote $\mathbf{z} = (z_1, \ldots, z_m)^{\mathsf{T}}$. Since all $\Omega_{B_i}$ are predetermined, e.g., from Label (2) for the problem of cell phone activity distribution recovery, and $z_i$ are known, reconstructing spatial field $\mathbf{f}$ from (1) is essentially solving a linear system of equations for $\mathbf{f}$, i.e.,

$$\mathbf{z} = \mathbf{Af},$$

where the matrix $\mathbf{A} \in \mathbb{R}^{m \times n}$ is given by

$$A_{ij} = \begin{cases} 1, & \text{if } \mathbf{p}_j \in \Omega_{B_i}, \\ 0, & \text{otherwise.} \end{cases} \tag{3}$$

Since $m \ll n$, i.e., the number of equations is far smaller than the number of the unknowns, reconstructing $f(\mathbf{p}_1), \ldots, f(\mathbf{p}_n)$ from $z_1, \ldots, z_m$ is essentially a sparse recovery problem.

Directly solving the linear system of Equation (1) is infeasible, as it is an underdetermined system which has an infinite number of solutions. However, the spatial property of $f$ can be utilized as constraints to make the sparse recovery problem feasible and has a unique solution. We observe that spatial data usually exhibit local continuity or correlation within domain $\Omega$. For example, in the problem of cell phone activity density recovery, the activity density of a certain location highly depends on the population and activity at that place, e.g., the downtown has more population and cell phone

activity than suburban residential areas. In addition, the underlying area functionality and the spatial distributions of human activity density are often slowly changing over the domain $\Omega$ rather than suddenly jumping between different subregions.

Therefore, we can formulate our constrained spatial sparse recovery problem as the following:

$$
\begin{aligned}
\underset{f}{\text{minimize}} \quad & \int_{\Omega} (\nabla^2 f)^2 \, d\mathbf{p}, \\
\text{subject to} \quad & \mathbf{z} = \mathbf{A}\mathbf{f}, \\
& \mathbf{f} \geq 0,
\end{aligned}
\tag{4}
$$

by taking into account the non-negative property and the local spatial continuity (smoothness) of $f$. $\nabla^2 f = \frac{\partial^2 f}{\partial x^2} + \frac{\partial^2 f}{\partial y^2}$ is the Laplacian of $f$, and is utilized to encourage local similarity and penalize the roughness of the spatial field $f$. It is worth noting that once $f$ is reconstructed, we have not only recovered the densities $\mathbf{f}$ at the square centers $\mathbf{p}_1, \ldots, \mathbf{p}_n$, but can also recover the density $f(\mathbf{p})$ of any point $\mathbf{p} \in \Omega$, e.g., between the centers of two neighboring grid squares, although such a fine-grained recovery may not be needed in every application.

To further improve the recovery performance, we can utilize additional external demographic or social features at each location. In the problem of cell phone activity density reconstruction, cell phone activities are often correlated with the underlying population density and social functionalities (e.g., the percentage of green area, the number of schools, the number of businesses/restaurants, the number of sport facilities, and the number of bus stops, etc.) of the considered regions.

Specifically, suppose $\mathbf{w}_j = (w_{j1}, \ldots, w_{jq})^{\mathsf{T}}$ represents the feature vector consisting of $q$ external feature values of square $j$. When $\mathbf{w}_j$ is available as additional input, we can estimate the spatial density data in square $j$ by

$$
f(\mathbf{p}_j) = f'(\mathbf{p}_j) + \mathbf{w}_j^{\mathsf{T}} \boldsymbol{\beta},
\tag{5}
$$

where $f'(\mathbf{p})$ is an underlying spatial field functional that preserves local spatial continuity, while $\mathbf{w}_j^{\mathsf{T}} \boldsymbol{\beta}$ is a linear regression part based on the attributes of square $\mathbf{p}_j$ that allows position-specific variation or jumps.

In the presence of attributes, we can formulate the constrained spatial sparse recovery problem as

$$
\begin{aligned}
\underset{f', \boldsymbol{\beta}}{\text{minimize}} \quad & \int_{\Omega} (\nabla^2 f')^2 \, d\mathbf{p}, \\
\text{subject to} \quad & f(\mathbf{p}_j) = f'(\mathbf{p}_j) + \mathbf{w}_j^{\mathsf{T}} \boldsymbol{\beta}, \quad j = 1, \ldots, n, \\
& \mathbf{z} = \mathbf{A}\mathbf{f}, \\
& \mathbf{f} \geq 0.
\end{aligned}
\tag{6}
$$

Once we get the spatial field $f'$ and $\boldsymbol{\beta}$, we can reconstruct $f(\mathbf{p}_j)$ for all the squares using (5). For example, we can calculate the cell phone activity at a specific place by the summation of an underlying smooth spatial field $f'(\mathbf{p}_j)$ and a linear regression of location attributes, where the add-on regression helps to model the jump between two subregions if the two regions are quite different and have distinct functionalities or attributes.

## 3. Patched Estimation and Spatial Spline Regression

In this section, we present some tentative solutions and then show their limitations in solving our constrained spatial sparse recovery problem.

### 3.1. Patched Piece-Wise Constant Estimation

In our problem, we only have access to the aggregated volumes $z_i$ at locations $\mathbf{p}_{B_i}$. To infer the fine-grained spatial distribution of $z_i$ over subregion $\Omega_{B_i}$ that covers the point $B_i$, a first intuitive

heuristic is estimating $f(\mathbf{p}_j)$ as the volume $z_i$ divided by its area by assuming the density is distributed uniformly:

$$\bar{f}(\mathbf{p}_j) = \frac{z_i}{|\Omega_{B_i}|}, \text{ for each } \mathbf{p}_j \in \Omega_{B_i}, \tag{7}$$

where $|\Omega_{B_i}|$ is the area of $\Omega_{B_i}$. This method gives us a patched piece-wise constant estimation. Note that we use *patch* to refer to $\Omega_{B_i}$ in this paper, which is the subregion covered $B_i$.

However, the patched estimation gives an oversimplified solution. The reconstructed spatial field $\bar{f}(\mathbf{p}_j)$ may have jumps on the borders of neighboring patches, which is far from smooth. In reality, the spatial field $f(\mathbf{p}_j)$ should change smoothly over the domain, as the underlying characteristics also change smoothly across different regions. Hence, $f(\mathbf{p}_j)$ should not be constant within each patch $\Omega_{B_i}$.

*3.2. Spatial Spline Regression*

Given the above observation, we can naturally come up with a second idea, which is learning a smooth estimation of $\bar{f}(\mathbf{p}_j)$ by spatial smoothing techniques. In the following, we introduce the powerful smoothing technique named Spatial Spline Regression (SSR) proposed in Sangalli et al. [16]. We will show how it can be applied to our particular spatial data reconstruction problem, as well as point out its limitations in solving the problem.

Given $l$ data points in $\Omega$, which contains the following information: (1) their positions $\{\mathbf{p}_j\}_{j=1}^l$; (2) the values of these $l$ points: $\{h_j\}_{j=1}^l$; and (3) their feature vectors $\{\mathbf{w}_j\}_{j=1}^l$, SSR is able to fit a smooth spatial field $f$ by minimizing the following equation [14,16], i.e.,

$$\underset{\beta, f}{\text{minimize}} \sum_{j=1}^l \left( h_j - \mathbf{w}_j^\mathsf{T} \beta - f(\mathbf{p}_j) \right)^2 + \lambda \int_\Omega (\nabla^2 f)^2 \, d\mathbf{p}, \tag{8}$$

where $f$ is assumed to be twice-differentiable over $\Omega$, and $\nabla^2 f = \frac{\partial^2 f}{\partial x^2} + \frac{\partial^2 f}{\partial y^2}$ denotes the *Laplacian* of $f$ to penalize the roughness of $f$. The hyper parameter $\lambda$ is used to trade the smoothness of $f$ off for a better approximation to data value $h_j$.

However, the challenge to solving problem (8) is that it involves searching for a functional $f$ over a possibly non-convex domain $\Omega$ that may have strong concavities, complicated boundaries, and even interior holes. Although kernel-based methods [26] are also a commonly used smoothing technique, their major drawback is that, by using uniformly damping weights in distance-based kernels, they tend to link data points across unrelated or weakly related subregions in an irregularly shaped non-convex domain.

We now briefly describe how spatial spline regression [16] can solve problem (8) via finite element analysis for any irregularly shaped domain $\Omega$. SSR splits a domain $\Omega$ by transforming it into a triangular mesh with triangulation methods (e.g., Delaunay triangulation [27]). After triangulation, it defines a polynomial function on each triangle, such that the summation of these polynomial functions defined on different pieces closely approximates the desired spatial field $f$.

Specifically, let $\zeta_1, \ldots, \zeta_K$ denote the vertices of all the small triangles, which are called control points and can be adaptively selected by available data points. Define a piecewise linear or quadratic basis function $\psi_k(x, y)$ called *Lagrangian finite element* with $(x, y) \in \Omega$, associated with each control point $\zeta_k$ such that $\psi_k$ evaluates to 1 at $\zeta_k$ and is equal to 0 at all other control points. Therefore, according to the *Lagrangian property of the basis*, we can approximate $f(x, y)$ for any $(x, y) \in \Omega$ only using the values of $f$ on the $K$ control points, i.e., $\mathbf{f}_K := (f(\zeta_1), \ldots, f(\zeta_K))^\mathsf{T}$. That is, if we let $\psi(x, y) := (\psi_1(x, y), \ldots, \psi_K(x, y))^\mathsf{T}$ denote the $K$ predefined basis functions, each corresponding to a control point, then we have

$$f(x, y) = \sum_{k=1}^K f(\zeta_k) \psi_k(x, y) = \mathbf{f}_K^\mathsf{T} \psi(x, y). \tag{9}$$

Since $\psi_1(x,y),\ldots,\psi_K(x,y)$ are predefined and known a priori, the variational estimation of $f$ in problem (8) boils down to the estimation of only $K$ scalar values, i.e., $\mathbf{f}_K = (f(\zeta_1),\ldots,f(\zeta_K))^\top$.

In fact, it is shown in Sangalli et al. [16] that with the piece-wise approximation given by (9), solving (8) is simply solving a set of linear equations for $\hat{f}(\zeta_1),\ldots,\hat{f}(\zeta_K)$. The estimator $\hat{f}(x,y)$ for $f$ can then be derived from (9) as

$$\hat{f}(x,y) = \hat{\mathbf{f}}_K^\top \psi(x,y).$$

It is worth noting that commodity triangulation software for finite element analysis is readily available in many free and commercial finite element packages. For example, Delaunay triangulations of a set of data location points (e.g., [27]) $V$ are such that no point in $V$ is inside the circumcircle of any triangle; they maximize the minimum angle of all the triangle angles, avoiding stretched triangles.

Now, we can see that if $l = n$ and we plug $h_j = \bar{f}(\mathbf{p}_j)$, $j = 1,\ldots,n$ into problem (8), we will get a new density surface $\hat{f}$ as a solution to the SSR problem (8) that is a smoothened approximation of the patched estimates $\bar{f}(\mathbf{p}_j)$.

However, SSR given by (8) can not accommodate any constraints, which is the major limitation in solving our problem. Specially, in our case, SSR does not enforce the aggregated volume constraint (1) (or $\mathbf{z} = \mathbf{Af}$ in (4)). Therefore, SSR gives no guarantee that the estimated densities in each patch $\Omega_{B_i}$ will sum up to the observed volume $z_i$ on the point $B_i$. In this way, SSR would likely cause large estimation errors as it violates the constraint.

## 4. An ADMM Algorithm for Constrained Spatial Smoothing

Our spatial sparse recovery problem (4) is different from (8) from two aspects: the additional constraints and the loss function. As a consequence, we can not directly apply the previous SSR method to solve it. A new approach is needed to handle our new loss function with constraints.

In this section, we propose to utilize the Alternating Direction Method of Multipliers (ADMM) [28], to decompose our constrained optimization problem into two sub-problems that can be solved effectively by SSR and Quadratic Programming (QP), respectively. Algorithm 1 presents the proposed ADMM algorithm to learn our model parameters.

---

**Algorithm 1:** Constrained Spatial Smoothing by ADMM

---

**Input:** The $m$ observed volume of base stations $\mathbf{z} = (z_1,\ldots,z_m)^\top$, smoothing parameter $\lambda$,
      penalty parameter $\beta$, initialize $\boldsymbol{\alpha} = \boldsymbol{\alpha}^0$, $\mathbf{f} = \mathbf{f}^0$.
**Output:** Spatial field and parameters $\hat{f}$, $\hat{\boldsymbol{\beta}}$. Estimation values on $n$ locations
      $\hat{\boldsymbol{f}} = (\hat{f}(\mathbf{p}_1),\ldots,\hat{f}(\mathbf{p}_n))$.
1: **for** $iter = 1,\ldots,$ maxIter **do**
2:     Update $f$ by solving (18) using Quadratic Programming.
3:     Update $g$ by solving (19) using Spatial Spline Regression.
4:     Update $\boldsymbol{\alpha}$ according to (17).
5: **end for**

---

First, we introduce the following indicator function $\mathbb{1}_{\mathbf{f}}$,

$$\mathbb{1}_{\mathbf{f}} = \begin{cases} 0, & \text{if } \mathbf{f} \geq 0 \text{ and } \mathbf{z} = \mathbf{Af}, \\ \infty, & \text{otherwise.} \end{cases} \tag{10}$$

With the indicator function, the original problem (4) is equivalent to

$$\underset{f}{\text{minimize}} \quad \lambda \int_\Omega (\nabla^2 f)^2 \, d\mathbf{p} + \mathbb{1}_{\mathbf{f}}, \tag{11}$$

where $\lambda$ is a hyper parameter that controls the smoothness of $f$.

Second, we introduce an auxiliary variable **g** that is defined as

$$\mathbf{g} := (g(\mathbf{p}_1), \dots, g(\mathbf{p}_n))^\mathsf{T}. \tag{12}$$

This variable is utilized to split the convex optimization problem into two sub-convex problems. With **g**, we can formulate the problem as the standard ADMM format,

$$\begin{aligned} \underset{f}{\text{minimize}} \quad & \lambda \int_\Omega (\nabla^2 f)^2 \, \mathrm{d}\mathbf{p} + \mathbb{1}_\mathbf{g}, \\ \text{subject to} \quad & \mathbf{f} = \mathbf{g}. \end{aligned} \tag{13}$$

The *augmented Lagrangian* for (13) is

$$\begin{aligned} \underset{}{\text{minimize}} \quad \mathcal{L}_\rho(\mathbf{f}, \mathbf{g}, \boldsymbol{\alpha}) = & \lambda \int_\Omega (\nabla^2 f)^2 \, \mathrm{d}\mathbf{p} + \mathbb{1}_\mathbf{g} \\ & + \boldsymbol{\alpha}^\mathsf{T}(\mathbf{g} - \mathbf{f}) + \frac{\rho}{2} \|\mathbf{g} - \mathbf{f}\|_2^2, \end{aligned} \tag{14}$$

where $\boldsymbol{\alpha} = (\alpha_1, \dots, \alpha_n)^\mathsf{T}$ is the dual variable, and $\rho > 0$ is the penalty parameter in ADMM. Then, the ADMM consists of the following iterations:

$$\mathbf{g}^{k+1} := \underset{\mathbf{g}}{\arg\min} \, \mathcal{L}_\rho(\mathbf{f}^k, \mathbf{g}, \boldsymbol{\alpha}^k), \tag{15}$$

$$\mathbf{f}^{k+1} := \underset{\mathbf{f}}{\arg\min} \, \mathcal{L}_\rho(\mathbf{f}, \mathbf{g}^{k+1}, \boldsymbol{\alpha}^k), \tag{16}$$

$$\boldsymbol{\alpha}^{k+1} := \boldsymbol{\alpha}^k + \rho(\mathbf{f} - \mathbf{g}). \tag{17}$$

For the **g**-update step in each iteration, Label (16) is equivalent to

$$\begin{aligned} \underset{\mathbf{g}}{\text{minimize}} \quad & \frac{\rho}{2} \|\mathbf{g}\|_2^2 + (\boldsymbol{\alpha}^\mathsf{T} - \rho \mathbf{f}^\mathsf{T}) \mathbf{g}, \\ \text{subject to} \quad & \mathbf{g} \geq 0, \\ & \mathbf{z} = \mathbf{A}\mathbf{g}. \end{aligned} \tag{18}$$

We can solve this convex problem efficiently by Quadratic Programming (QP).

For the **f**-update step in each iteration, Equation (15) is equivalent to

$$\underset{f}{\text{minimize}} \quad \left\| \left( \boldsymbol{\alpha}^\mathsf{T} + \rho \mathbf{g}^\mathsf{T} \right) / 2 - \mathbf{f} \right\|_2^2 + \lambda \int_\Omega (\nabla^2 f)^2 \, \mathrm{d}\mathbf{p}, \tag{19}$$

which is exactly the form of (8) with $h_j = \left( \alpha_j + \rho g(\mathbf{p}_j) \right) / 2$ and $w_j = 0$, thus can be solved efficiently by SSR. It should be noted that $\lambda$ is the penalty parameter which controls the smoothness of $f$. If it is small, we put little emphasis on the smoothness, and the estimated surface $f$ will be over fitted. If it is too big, the surface will be too smooth, which can cause underfitting.

For the case with attributes, the algorithm does not require major changes. We just need to replace **f** by $\mathbf{f} + \mathbf{W}\boldsymbol{\beta}$ in (19), where $\mathbf{W} := (\mathbf{w}_1, \dots, \mathbf{w}_n)^\mathsf{T}$ represents the attributes and $\boldsymbol{\beta}$ is the corresponding contributions.

Our proposed ADMM training algorithm is able to efficiently reconstruct the spatial field and fit the covariates for our constrained spatial sparse recovery problem. In **g**-update step, it enforces the constraints by solving a constrained QP with no need to worry about smoothing; in a **f**-update step, it approximates the obtained **g** with a smooth $f$ using the SSR-based smoothing technique. In this way, we decouple the handling of smoothing and constraints which was not possible in pure SSR previously.

## 5. Performance Evaluation

In this section, we perform an extensive case study of the approach we described above in order to demonstrate its applicability. We picked the cell phone data as an example of how the model can solve empirical problem and compare the model's performance to other approaches.

### 5.1. Dataset Description

The model in (13) is general and not attached to any particular empirical problem, and it does not contain many implicit assumptions. However, in order to measure its performance, we evaluate the model using real-world data. Due to generality of the proposed learning algorithm, the range of possible data sets is potentially big. For our empirical case study, we chose cell phone data, where there exists a problem of recovering a spatial field from coarse aggregations observed at sparse cell phone towers. We do not overestimate the problem, but rather see this particular data set suitable for an extensive case study.

The Milan Call Description Records (CDR) dataset is a part of the Telecom Italia Big Data Challenge dataset provided by Telecom Italia Mobile. It contains the telecommunications activity records from 1 November 2013 to 31 December 2013 in the city of Milan [25]. The dataset divides Milan into a $100 \times 100$ square grid, where each square is size of about 235 m $\times$ 235 m. In the dataset, each record consists of six entries: square ID, incoming call activity, outgoing call activity, incoming SMS activity, outgoing SMS activity, and time-stamp of 10-minute time slot. The values of the four types of activities are normalized to the same scale.

Another dataset we utilized is the Milan geographical attribute dataset available from the Municipality of Milan's Open Data website [1]. This dataset consist of features of central 2726 squares among the whole 10,000 squares. The features of each square include: population, green area percentage, number of sport centers, number of universities, number of businesses, and number of bus stops. Figure 1 shows the area covered by these grid squares. The 2726 squares covers the central part of the Milan city and contains the majority of telecommunication activities in the dataset. We refer to [2] for more detailed description about this dataset. In our experiments, we compare the performance of different approaches on these squares.

The general problem of recovering a spatial field from coarse aggregations observed at sparse points in the field in this particular case study is reformulated into the problem of recovering the distribution of cell phone activities over the whole 2726 square regions given that only aggregated activity observations in base stations are known. We need to further process the Milan CDR dataset to study this problem.

First, we sum up the four types of activities during 1 November 2013 to 28 November 2013 and 1 December 2013 to 28 December 2013, respectively, to come up with the activity volume of each squares during November or December. These two datasets are served as the ground-truth datasets of Milan cell phone activity distributions. Figure 1a,b show the heat maps of activity volumes in each square during November and December.

Second, after we aggregated the two months' activities for each square, we need to set the locations of base stations (BSs). According to [29], there are roughly 200 base stations in Milan. However, the exact locations are not available. Thus, we assume the 200 BSs are randomly distributed according to the following probability distribution

$$\Pr(\text{Set square } i \text{ as BS}) = f(\mathbf{p}_i) / \sum_{j=1}^{N} f(\mathbf{p}_j), \tag{20}$$

where $f(\mathbf{p}_i)$ is the cell phone activity volume in square $i$, $i = \{1, \ldots, N\}$, $N = 2726$ is the number of squares we are focusing on. Notice that, when we have 200 base station's aggregated observations, they only cover 7.34% of the whole 2726 squares region. This is extremely sparse and makes our problem highly challenging. In addition, we also assume $n_{\text{BS}} = 100$ and choose 100 squares as BSs according

to the same probability distribution to stress-test our algorithm's capability under even sparser observations. Figure 2a,b show the base station distributions for $n_{BS} = 200$ and $n_{BS} = 100$, respectively.

After sampling the location of base stations, for each square, we assign the activity of it to its closest base station. When multiple base stations are equidistant from a square, the activity of the square will be evenly distributed among these base stations. We then assume we only know the aggregated activities in base station squares, which is usually the true case in reality. Figure 2c shows the regions split by 100 base stations, where each colour patch is a region charged by one base station. To save space, we don't present the figure for 200 base stations.

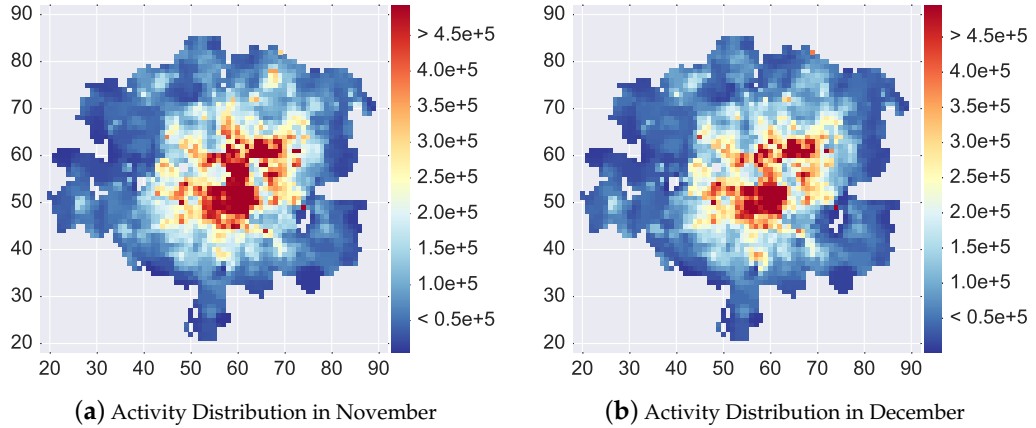

(**a**) Activity Distribution in November      (**b**) Activity Distribution in December

**Figure 1.** The cell phone activity distributions of Milan. It shows the metropolitan area of Milan, Italy, and the area covered by the 2726 grid squares. (**a**,**b**) show the heat map of cell phone activities (Call + SMS) during November and December respectively.

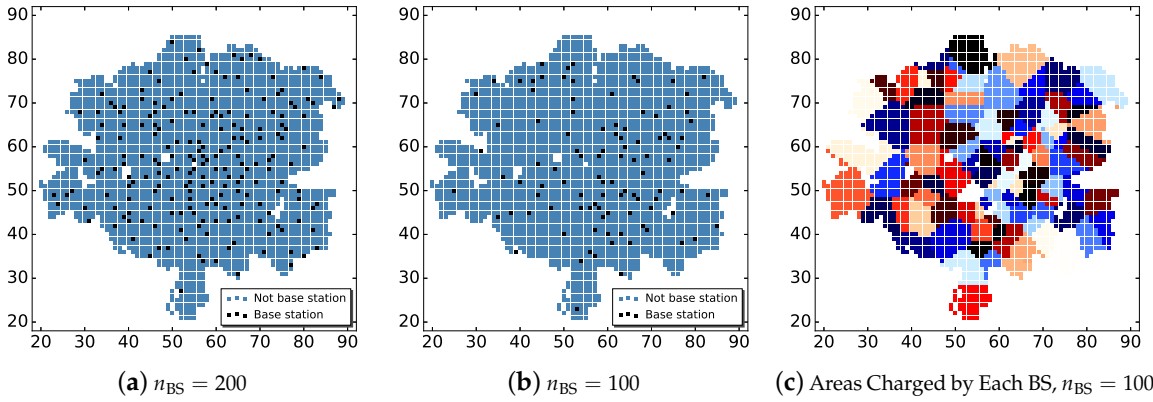

(**a**) $n_{BS} = 200$      (**b**) $n_{BS} = 100$      (**c**) Areas Charged by Each BS, $n_{BS} = 100$

**Figure 2.** The location distributions of sampled base stations and the areas in charged by them. (**a**,**b**) shows the sampled base station distributions for $n_{BS} = 200$ and $n_{BS} = 100$; (**c**) shows the areas in charged by different base stations for $n_{BS} = 100$.

*5.2. Experimental Setup*

Algorithms Evaluated

We test our proposed approach and compare it with three baseline methods. In particular, we evaluate and compare the following models using the aggregated November and December datasets, with number of base stations $n_{BS} = 200$ or $n_{BS} = 100$ for stress testing.

- **Patched Estimation**: estimate the cell phone activity distribution by patched piece-wise constant estimation, that is, assume cell phone activity density is distributed uniformly within each

sub-region $\Omega_{B_i}$, i.e., the area covered by base station $B_i$, and estimate each square's activity volume by (7).

- **Patched Estimation + SSR 1**: first estimate *only base station* activity volumes by (7). Use these sparse points to fit a smooth surface by running Spatial Spline Regression to obtain the estimated cell phone activity in all squares.

- **Patched Estimation + SSR 2**: as opposed to the previous model, get the initial estimation of the activity volume of *all squares* by Patched Estimation. Then, use all these points to fit a smooth surface by running Spatial Spline Regression to obtain the final estimated cell phone activity in all squares.

- **Constrained Spatial Smoothing**: first get the initial estimation of the activity volume of all squares by Patched Estimation, then run Constrained Spatial Smoothing algorithm to get the final activity volumes estimation of all squares.

- **Constrained Spatial Smoothing + Features**: in this case, we incorporate the geographical features into Constrained Spatial Smoothing algorithm.

We set the penalty parameter $\lambda = 1$ when $n_{BS} = 200$ and $\lambda = 10$ when $n_{BS} = 100$, for all methods that utilizes SSR. The geographical features of Milan are only incorporated in the last algorithm described above. In addition, for the implementation of Spatial Spline Regression, we use the *fdaPDE* R Package [30].

To compare different approaches, we evaluate the performance by the Mean Relative Error (MRE) of the produced activity estimates for the true activity values. The relative error of an estimation $\hat{f}(\mathbf{p}_j)$ compared to the true value $f(\mathbf{p}_j)$ is defined as $|\hat{f}(\mathbf{p}_j) - f(\mathbf{p}_j)|/f(\mathbf{p}_j)$.

### 5.3. Performance Evaluation

### 5.3.1. Comparison of Different Algorithms

We show the cumulative distribution function (CDF) of Relative Errors given by different approaches in Figures 3 and 4. In addition, we compare the estimation's Mean Relative Error of different approaches in Figure 5. It is quite clear that our proposed algorithms outperform other three baseline approaches significantly in all cases ($n_{BS} = 200$ and $n_{BS} = 100$, data aggregated in November and in December).

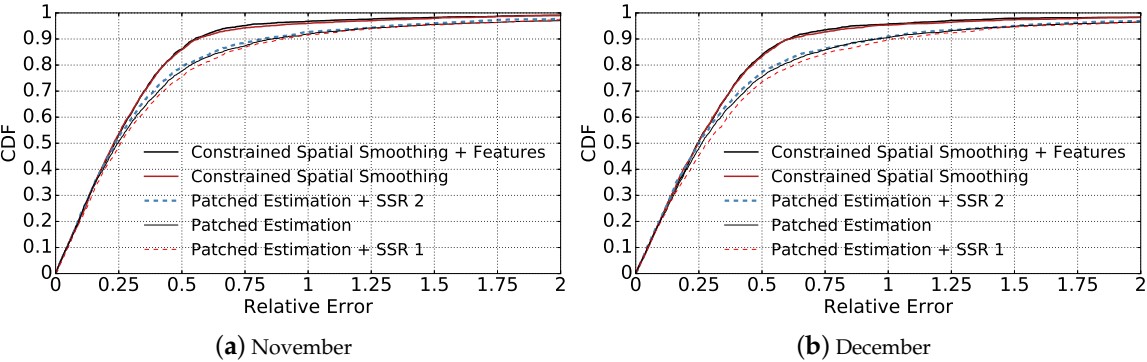

(**a**) November        (**b**) December

**Figure 3.** Comparison of the CDFs of estimation relative errors given by different methods when $n_{BS} = 200$. The legends follow the same order as the curves at relative error = 0.5. (**a**) compares the CDFs based on the data aggregated in November; (**b**) compares the CDFs based on the data aggregated in December.

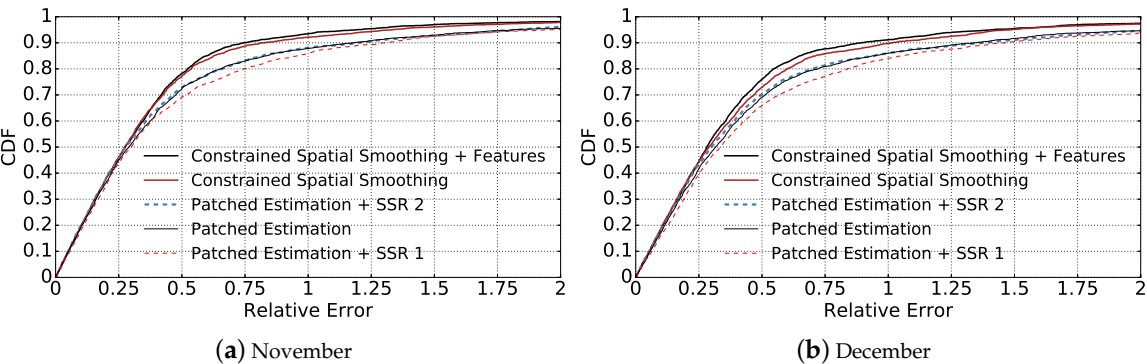

**Figure 4.** Comparison of the CDFs of estimation relative errors given by different methods when $n_{BS} = 100$ for stress-testing. The legends follow the same order as the curves at relative error = 0.5. (**a**) compares the CDFs based on the data aggregated in November; (**b**) compares the CDFs based on the data aggregated in December.

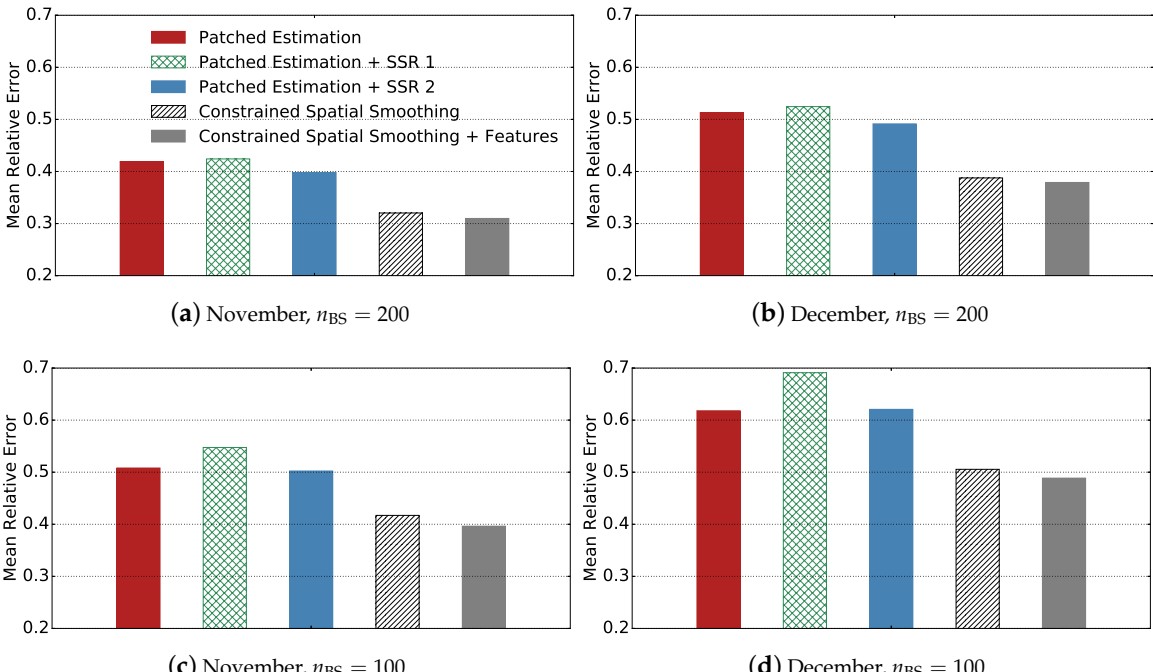

**Figure 5.** Comparison of the estimation's Mean Relative Error of different methods when $n_{BS} = 200$ or $n_{BS} = 100$ for stress-testing. In each figure, the bars from left to right stands for Patched Estimation, Patched Estimation + SSR 1, Patched Estimation + SSR 2, Constrained Spatial Smoothing, and Constrained Spatial Smoothing + Features respectively. (**a**) we use the data aggregated in November, and set number of base stations to be 200, similarly for (**b–d**).

By comparing Patched Estimation + SSR 1 with Patched Estimation approach, we can see that using spatial smoothing based on only base station squares' observations leads to worse performance than patched estimation. This can be explained by the smoothing property of SSR and how we set the values of base station squares. As we described, we set the activity value of base stations by averaging the total activity amount of each base station on all squares it covers. Thus, given the activity $\frac{z_i}{|\Omega_{B_i}|}$, ($|\Omega_{B_i}|$ denotes the number of squares within region $\Omega_{B_i}$) of a base station $B_i$, the true activities of itself and its surrounding squares within region $B_i$ are distributed with a mean of $\frac{z_i}{|\Omega_{B_i}|}$. Given two base stations $B_1$ and $B_2$ that are close to each other, with aggregated activities of $z_1$ and $z_2$ respectively, the Spatial Smoothing approach will fit a smooth surface between the two base stations. Suppose

$z_1 > z_2$, in this case, overall, the activities of $B_1$'s neighbour squares will be underestimated, and that of $B_2$ will be overestimated. Therefore, Patched Estimation + SSR 1's performance is not as good as Patched Estimation.

By comparing Patched Estimation + SSR 2 with Patched Estimation and Patched Estimation + SSR 1, we can observe that applying spatial smoothing on the results of patched estimation improves the performance. This proves the rationality and effectiveness of introducing smoothness into the estimated cell phone activity distribution surface.

Our proposed approach achieves much better performance compared with the three baseline methods. By using Constrained Spatial Smoothing instead of applying Spatial Spline Regression directly, we are able to fit a smooth activity distribution while forcing it to match the observations of base station squares (the aggregated activity volumes) at the same time. By comparing Constrained Spatial Smoothing that incorporates additional features of each square with the version without features, we can see that the performance is further improved. The reason is that the heterogeneity of different locations will influence the telecommunication activity distribution, therefore making the distribution not smooth everywhere. Incorporating additional features into our model can help to explain the residuals between estimated smooth distribution and the true activity distribution, therefore further increasing estimation accuracy. By comparing Figure 3 and Figure 4, we also can see that incorporating additional features into Constrained Spatial Smoothing becomes more important when the base stations are more sparse.

The performance of different methods on the December dataset is worse than on the November dataset. This is because there are multiple holidays in December. The cell phone activities will become more irregular than usual during holidays, as discussed in Cici et al. [2] and Ratti et al. [29].

Figure 6a–c show the distribution surfaces of true cell phone activity volumes, estimated volumes by Patched Estimation, and estimated volumes by Constrained Spatial Smoothing with features when $n_{\text{BS}} = 200$ using the November dataset. We can see that the Patched Estimation approach fits a stepped surface, while our approach gives a much smoother surface.

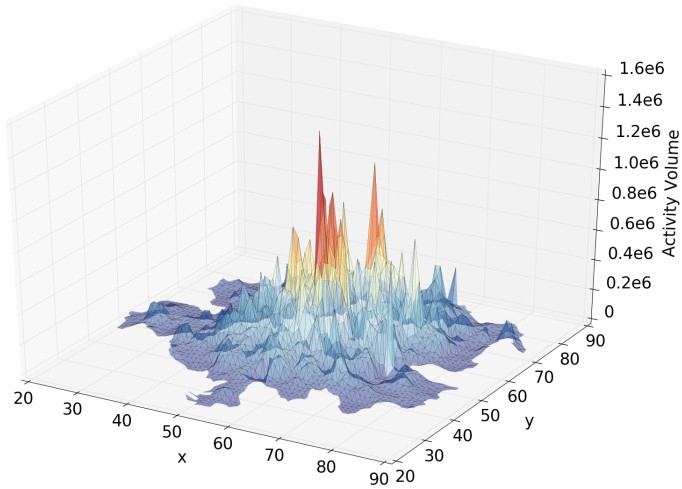

(**a**) Real cell phone activity distribution.

**Figure 6.** *Cont.*

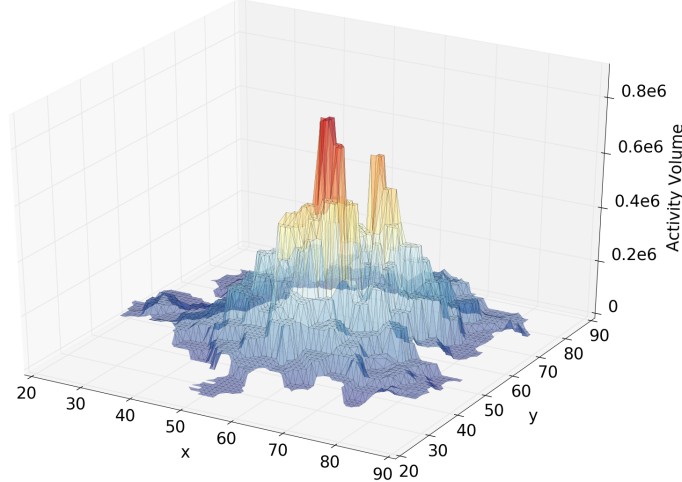

(**b**) Estimated cell phone activity distribution by Patched Estimation.

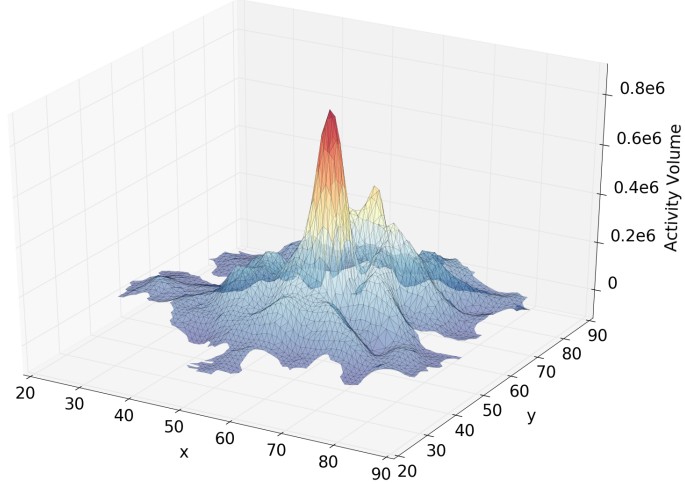

(**c**) Estimated cell phone activity distribution by Constrained Spatial Smoothing + Features.

**Figure 6.** Comparison of the activity distributions. (**a**) real cell phone activity distribution; (**b**) estimated distribution by Patched Estimation method; (**c**) estimated distribution by our method.

For time efficiency, experiments based on the Milan Call Description Records (CDR) dataset show that the average time for our approach to converge is less than five minutes on a MacBook Pro with a 2 GHz Intel Core i7 processor, and 8 GB memory. This proves that our system is highly efficient and practical.

5.3.2. Impact of Smooth Penalty Parameter $\lambda$

Figure 7 shows how the the estimation's Mean Relative Error varies when $\lambda$ increases from $10^{-4}$ to $10^3$. We make two interesting observations. First, $\lambda$ around 1~10 usually gives the best performance. Too big or too small $\lambda$ will decrease the estimation accuracy. This is reasonable, as when $\lambda$ is too small, we put little emphasis on the smoothness of estimated surface, thus the performance will suffer. If $\lambda$ is too big, it enforces a smooth surface, which also doesn't match the reality. Second, when we have less base stations, $\lambda$ that gives the best performance will increase (from 1 to 10). In addition, we can see that the performance of the model with $\lambda$ between 1~100 does not significantly change when $n_{\text{BS}} = 100$. That indicates the following: when the base station distribution is more sparse,

the estimation performance is less sensitive to $\lambda$ when it is around the best value (1 for $n_{BS} = 200$ and 10 for $n_{BS} = 100$).

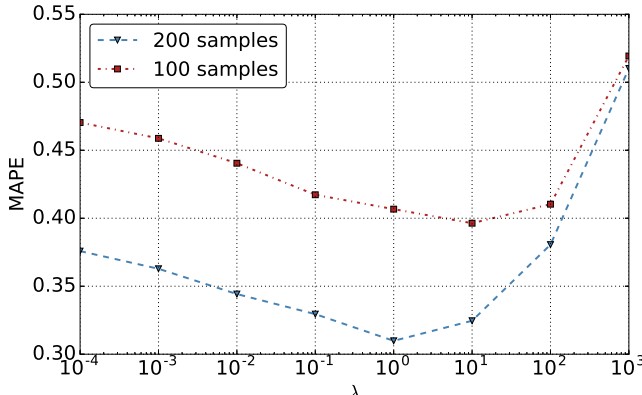

**Figure 7.** Influence of $\lambda$ to estimation's Mean Relative Error when $n_{BS} = 200$ and $n_{BS} = 100$ for stress-testing. The figure is based on the November dataset. Results on the December dataset are similar.

## 6. Related Work

The Telecom Italia Big Data Challenge dataset is widely used to study different problems [2–5,20–24]. However, little research work has been done to estimate the spatial distribution of cellphone activity itself, despite the great value of this problem.

There are various tasks where the key problem is estimating a spatial field over a region based on observations of sampled points, such as house price estimation and population density estimation. Chopra et al. [26] model the underlying surface of land desirability using kernel-based interpolation. However, it is hard to choose the form of kernel functions and tune a large number of hyper-parameters. Spatial Spline Regression technique is applied to the problem of population density estimation in Sangalli et al. [16]. However, in our problem, we only get the accumulated activity density in base stations, rather than real densities in each base station location. In addition, BS locations distribution is highly sparse in our case.

Although a range of kernel-based methods [26,31,32] can be applied to fit a spatial field, the common drawback of these approaches is that, by using uniformly damping weights in distance-based kernels, they tend to link weakly related data points across areas in a non-convex domain. Spatial spline regression [16] on the other hand uses finite-element analysis approach to jointly solve for $f$ and $\beta$ from the model described by Equation (8) over any irregularly shaped domain $\Omega$.

As it was discussed earlier, the fine-grained data for the distribution of the volume of calls and SMS are not usually available. A common type of data is the data collected by cell phone base stations. Sometimes, cell phone providers interpolate the data collected by the base stations as is discussed in Manfredini et al. [33]. Some researchers interpolate the data to obtain fine grained distributions as in Ratti et al. [29]. However, in Ratti et al. [29], authors do not evaluate the performance of the interpolated distribution. To the best of our knowledge, there is no extensive work done in trying to obtain optimal reconstructions of fine grained cell phone data distribution. We are the first to apply the latest spatial functional analysis techniques to cellphone activity distribution modeling, assuming the activity densities consist of a regression part based on social or demographical statistic features and a spatial field that captures the underlying smoothness property of cellphone activities. In particular, we leverage the idea of spatial spline regression to handle any irregularly shaped geographic regions. We have developed a novel Constrained Spatial Smoothing approach and corresponding training algorithm to recover spatial distribution of cellphone activities from highly sparse observations.

## 7. Conclusions

In this paper, we study the problem of inferring the fine-grained spatial distribution of certain density data in a region based on the aggregate observations recorded for each of its subregions, which is extremely challenging and seldom visited before, and analyze the challenges of it. We propose the Constrained Spatial Smoothing (CSS) approach that exploits both the intrinsic smooth property of underlying factors and the additional features from external social or domestic statics. We further propose a training algorithm which combines the Spatial Spline Regression (SSR) technique and ADMM technique to learn our model parameters efficiently.

To evaluate our algorithm and compare it with various other approaches, we run extensive evaluations based on the Milan Call Detail Records dataset provided by Telecom Italia Mobile. The simulation results on the dataset show that our algorithm significantly outperforms other baseline approaches by a great percentage. (Note that cross validation and statistical testing are techniques that are usually applied in experiments. However, both techniques require sampling effectively from the sparse spatial data while keeping the intrinsic spatial structure, which is difficult in our problem.) This shows that jointly modeling the underlying spatial continuity and the local features that characterize the heterogeneity of different locations can effectively improve the performance of spatial recovery.

Although we use the data on cell phone activities to illustrate our methodology, our algorithm is not limited to solving the problem of inferring the distribution of cell phone activities, but is also applicable to a variety of problems where estimating an implicit or explicit smooth surface is required, such as inferring the spatial distribution of population densities based on the aggregate population observed at sparsely scattered polling stations, reconstructing a fine-grained geographical distribution of users for an Internet media provider or retailer only from aggregated user counts observed at certain datacenters or points of presence (PoPs), and so on.

**Author Contributions:** Conceptualization, B.L., B.M., L.K. and D.N.; Data curation, B.L. and B.M.; Formal analysis, B.L., B.M., L.K. and D.N.; Investigation, B.L., B.M., L.K. and D.N.; Methodology, B.L., B.M., L.K. and D.N.; Project administration, B.L., B.M., L.K. and D.N.; Software, B.L. and B.M.; Supervision, B.L., B.M., L.K. and D.N.; Validation, B.L., B.M., L.K. and D.N.; Visualization, B.L., B.M. and L.K.; Writing—original draft, B.L., B.M., L.K. and D.N.; Writing—review and editing, B.L., B.M. and D.N.

**Funding:** This research received no external funding.

**Conflicts of Interest:** The authors declare no conflict of interest.

## Abbreviations

The following abbreviations are used in this manuscript:

CSS     Constrained Spatial Smoothing
SSR     Spatial Spline Regression
SMS     Short Message Service
ADMM    Alternating Direction Method of Multipliers
QP      Quadratic Programming
CDR     Call Detail Records
PoPs    Presence of Points
CDF     Cumulative Distribution Function

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
