# Peer review of "Spatial Data Reconstruction via ADMM and Spatial Spline Regression†"

_applsci, doi:10.3390/app9091733_

Round 1
Reviewer 1 Report
I have reviewed the manuscript applsci-481840 and I believe the authors have carried out an interesting study to reconstruct fine-grained spatial densities from coarse-grained measurements. Nevertheless, this study is no new. The authors advised that this paper is an extended version of their own paper published in the 2017 IEEE International Conference on Data Mining (ICDM). I have consulted the referred paper, titled “Recover Fine-Grained Spatial Data from Coarse Aggregation” and, in fact, much of this paper is included in the manuscript applsci-481840, including text and figures. I consider this manuscript does not comply with what is specified in the “Publication Ethics Statement” section of the Applied Sciences Journal. Because of this, I consider this article is not suitable for publication in Applied Sciences Journal.
Regarding to the manuscript, there are some suggestions and comments I detail below:
Major comments:
1. Abstract section: The abstract presents a strange structure. Remember that the abstract should follow this style: 1) Background; 2) Methods; 3) Results and 4) Conclusion. In the manuscript, when the authors want to explain the study, they show the case of study as an example (L 16-17). Later, near the end of the abstract (L 24-25), the authors explain the origin of the data used on the case of study. Additionally, the results are practically nonexistent and there is no discussion.
2. L 34-37. Introduction section: The beginning of the sentence is not very formal for this type of publication. If the authors want to referrer different papers related to the background of the study, the beginning should be more formal.
3. L 47-49. Introduction section: this sentence is showing the objective of the paper, and should be placed near the end of the Introduction section.
4. L 58-62. Introduction section: In two consecutive sentences, the references [12-16] are shown. Nevertheless, in any of the sentences the name of the smoothing techniques are named.
5. L 67-82. Introduction section: I think this paragraph is part of the methodology proposed, and it should be located in other section of the manuscript, but not in the Introduction section.
6. L 83-93. Introduction section: I think this paragraph is part of the data used, and it should be located in other section of the manuscript, but not in the Introduction section.
7. L 94-98. Introduction section: I think this paragraph is part of the results, and it should be located in other section of the manuscript, but not in the Introduction section.
8. The discussion of the results is not very clear. In consider that part of the information shown in the 6th section “Related work”, should be placed in the introduction section (especially the first paragraph).
Minor comments:
1. Be careful with the use of bold and italic words. The authors have copied the enhancement letter of the previous publication (2017 IEEE International Conference on Data Mining (ICDM)), but the Applied Sciences Journal has other rules.
2. Be careful with the heading of the 3rd and 4th sections. The heading is divided in two lines. The authors have copied the enhancement letter of the previous publication (2017 IEEE International Conference on Data Mining (ICDM)).
3. The captions of the Figures 1 and 2 should be revised. If the authors show three different pictures, maps or draws (a, b and c) in one figure, is because they have something in common. The captions of both, 1 and 2 figures should show a main description of the common idea and, later, the specific information of each one.
4. The captions of the Figures 3, 4 and 5 should be revised. In these figures, the main idea of the figures are shown very clearly, but the captions of the figures do not describe the difference among graphs: a and b in Figures 3 and 4, and a, b, c and d in Figure 5.
5. Is it necessary to divide the graphs shown in Figures 6, 7 and 8 in three different Figures?. Can all these three figures be grouped in one to compare more easily?.
6. When the reference is part of the sentence, you should indicate the name of the first author et al. Revise the text. Example of that: L392 (… as is discussed in [34]); L 393 (… as in [24]. However in [24] authors…)…
7. L 389. The reference "8" should be placed in square brackets [ ].
8. Between L 275 - 276 there are some sentences without number of lines.
Author Response
Response to Reviewer 1 Comments
Major comments:
Point 1: Abstract section: The abstract presents a strange structure. Remember that the abstract should follow this style: 1) Background; 2) Methods; 3) Results and 4) Conclusion. In the manuscript, when the authors want to explain the study, they show the case of study as an example (L 16-17). Later, near the end of the abstract (L 24-25), the authors explain the origin of the data used on the case of study. Additionally, the results are practically nonexistent and there is no discussion.
Response 1: Thanks a lot for the suggestion. We revised the abstract of the paper to make it more well-structured, as well as added more insights.
Point 2: L 34-37. Introduction section: The beginning of the sentence is not very formal for this type of publication. If the authors want to referrer different papers related to the background of the study, the beginning should be more formal.
Response 2: Thanks for suggestion. We have revised the first paragraph of the introduction to make it more formal.
Point 3: L 47-49. Introduction section: this sentence is showing the objective of the paper, and should be placed near the end of the Introduction section.
Response 3: The first sentence of the second paragraph in introduction is used to describe our problem. Then we illustrate the challenges in solving this problem. That is why we put this sentence here.
Point 4: L 58-62. Introduction section: In two consecutive sentences, the references [12-16] are shown. Nevertheless, in any of the sentences the name of the smoothing techniques are named.
Response 4: We have revised the sentences to show the names of different smoothing techniques.
Point 5: L 67-82. Introduction section: I think this paragraph is part of the methodology proposed, and it should be located in other section of the manuscript, but not in the Introduction section.
Response 5: We have reduced the math notations in the third paragraph of introduction. This paragraph serves to briefly summarize our main ideas in our algorithm. Therefore, we put it in introduction to show our contribution. In the methodology section, we describe our method in more details.
Point 6: L 83-93. Introduction section: I think this paragraph is part of the data used, and it should be located in other section of the manuscript, but not in the Introduction section.
Response 6: This paragraph is aiming to further illustrate our contribution in this paper. We put more detailed introduction about the dataset and experiments in Section 5.
Point 7: L 94-98. Introduction section: I think this paragraph is part of the results, and it should be located in other section of the manuscript, but not in the Introduction section.
Response 7: Similarly, this paragraph is used to show the performance of our algorithm. Basically, in introduction, our logic flow is: first, introduce and define the problem; second, illustrate the challenges and the limits in existing approaches; third, briefly summarize our ideas to solve the problem; fourth, briefly introduce our experiments and results to show the effectiveness; and last, the paper organization.
Point 8: The discussion of the results is not very clear. In consider that part of the information shown in the 6th section “Related work”, should be placed in the introduction section (especially the first paragraph).
Response 8: Thanks for the suggestion, we have put the first part of the information in Section 6 into introduction to introduce the detailed information of the dataset. We have also re-organized the paragraphs that introduce the dataset and experimental results in introduction to make that section more well-structured.
Minor comments:
Point 1: Be careful with the use of bold and italic words. The authors have copied the enhancement letter of the previous publication (2017 IEEE International Conference on Data Mining (ICDM)), but the Applied Sciences Journal has other rules.
Response 1: Thanks for remind. We have checked the instructions for authors, and it seems that the use of bold and italic words in our manuscript is appropriate.
Point 2: Be careful with the heading of the 3rd and 4th sections. The heading is divided in two lines. The authors have copied the enhancement letter of the previous publication (2017 IEEE International Conference on Data Mining (ICDM)).
Response 2: We have fixed this issue.
Point 3: The captions of the Figures 1 and 2 should be revised. If the authors show three different pictures, maps or draws (a, b and c) in one figure, is because they have something in common. The captions of both, 1 and 2 figures should show a main description of the common idea and, later, the specific information of each one.
Response 3: We have revised the captions of Fig. 1 and 2 to show a main description of the common ideas first.
Point 4: The captions of the Figures 3, 4 and 5 should be revised. In these figures, the main idea of the figures are shown very clearly, but the captions of the figures do not describe the difference among graphs: a and b in Figures 3 and 4, and a, b, c and d in Figure 5.
Response 4: We have revised the captions of Figure 3, 4 and 5 to describe the difference among sub-figures more clearly.
Point 5: Is it necessary to divide the graphs shown in Figures 6, 7 and 8 in three different Figures? Can all these three figures be grouped in one to compare more easily?
Response 5: We have put Figure 6, 7, 8 into one figure (Figure 6) to compare different spatial fields.
Point 6: When the reference is part of the sentence, you should indicate the name of the first author et al. Revise the text. Example of that: L392 (… as is discussed in [34]); L 393 (… as in [24]. However in [24] authors…)…
Response 6: We have added the name of the first author et al. when the reference is part of the sentence.
Point 7: L 389. The reference "8" should be placed in square brackets [ ].
Response 7: This 8 denotes the equation 8. We have revised the writing to make it more clear.
Point 8: Between L 275 - 276 there are some sentences without number of lines.
Response 8: This is caused by equations. We have fixed all the similar problems in our LaTeX code.

Reviewer 2 Report
The authors study the problem of reconstructing fine-grained spatial densities from coarse-grained measurements.
Specifically, they study the problem of inferring the fine-grained spatial distribution of density data based on the observations recorded for each of sub-regions.
They propose a novel Constrained Spatial Smoothing for the problem of spatial data reconstruction.
The authors provide a training algorithm which combines the Spatial Spline Regression technique and propose ADMM technique to learn their model parameters efficiently.
Moreover, they applied extensive evaluations based on a large dataset of Call Detail Records and a geographical attribute data set from the city of Milan. Experimental results outperforms various state-of-the-art approaches.
The paper is overall well written and easy to follow.
The examined topic is of great interest and applicability. The provided analysis in is concrete and correct.
-A simple minor comment, in line 170 the title of the section is broken.
Author Response
Point 1:A simple minor comment, in line 170 the title of the section is broken.
Response:Thanks for point out this! We have fixed this problem in our new version.
Reviewer 3 Report
A kind of cross validation should be used for the experiments.
A statistical test should be used for the comparison of the examined methods.
The authors should explain why the proposed methodology seems to work well and present information about the time efficiency of their method.
Author Response
Response to Reviewer 3 Comments
Point 1: A kind of cross validation should be used for the experiments.
Response 1: Thank you for your suggestion. We agree that it will be great to do cross validation. However, the barrier is how to sample effectively from the sparse spatial data while keep the intrinsic spatial structure. We are not aware of any sophisticated method to fulfill that task. We added a remark as footnote in the concluding remarks.
Point 2: A statistical test should be used for the comparison of the examined methods.
Response 2: To conduct statistical testing, either theoretic properties need to be developed or bootstrapping can be used. The former is out of the scope of this manuscripts. The later again requires sophisticated sampling methods for sparse spatial data. We added a remark as footnote in the concluding remarks.
Point 3: The authors should explain why the proposed methodology seems to work well and present information about the time efficiency of their method.
Response 3: Jointly modeling the underlying spatial continuity and the local features that characterizing the heterogeneity of different locations can help improving the performance of spatial recovery. We revised our abstract and conclusion to explain why our methodology works well. For time efficiency, experiments based on the Milan Call Description Records (CDR) dataset show that the average time for our approach to converge is less than a few minutes on a MacBook Pro with a 2 GHz Intel Core i7 processor, and 8 GB memory. This proves that our system is highly efficient and practical. We added this information in the Section 5.

Round 2
Reviewer 1 Report
Thanks for the efforts to improve the manuscript.